# Novel Arylpiperazine Derivatives of Salicylamide with α_1_-Adrenolytic Properties Showed Antiarrhythmic and Hypotensive Properties in Rats

**DOI:** 10.3390/ijms24010293

**Published:** 2022-12-24

**Authors:** Elżbieta Żmudzka, Klaudia Lustyk, Agata Siwek, Małgorzata Wolak, Adam Gałuszka, Jolanta Jaśkowska, Marcin Kołaczkowski, Jacek Sapa, Karolina Pytka

**Affiliations:** 1Department of Social Pharmacy, Faculty of Pharmacy, Jagiellonian University Medical College, Medyczna 9, 30-688 Krakow, Poland; 2Department of Pharmacodynamics, Faculty of Pharmacy, Jagiellonian University Medical College, Medyczna 9, 30-688 Krakow, Poland; 3Department of Pharmacobiology, Faculty of Pharmacy, Jagiellonian University Medical College, Medyczna 9, 30-688 Krakow, Poland; 4Department of Automatic Control and Robotics, Silesian University of Technology, Akademicka 2A, 44-100 Gliwice, Poland; 5Department of Organic Chemistry and Technology, Faculty of Chemical and Engineering and Technology, Cracow University of Technology, Warszawska 24, 31-155 Krakow, Poland; 6Department of Medicinal Chemistry, Faculty of Pharmacy, Jagiellonian University Medical College, Medyczna 9, 30-688 Krakow, Poland

**Keywords:** alpha-1 adrenergic receptor, arylpiperazine derivatives, antiarrhythmic agents, hypotensive agents, statistical analysis, neural network, parameter identification

## Abstract

Cardiovascular diseases remain one of the leading causes of death worldwide. Unfortunately, the available pharmacotherapeutic options have limited effectiveness. Therefore, developing new drug candidates remains very important. We selected six novel arylpiperazine alkyl derivatives of salicylamide to investigate their cardiovascular effects. Having in mind the beneficial role of α_1_-adrenergic receptors in restoring sinus rhythm and regulating blood pressure, first, using radioligand binding assays, we evaluated the affinity of the tested compounds for α-adrenergic receptors. Our experiments revealed their high to moderate affinity for α_1_- but not α_2_-adrenoceptors. Next, we aimed to determine the antiarrhythmic potential of novel derivatives in rat models of arrhythmia induced by adrenaline, calcium chloride, or aconitine. All compounds showed potent prophylactic antiarrhythmic activity in the adrenaline-induced arrhythmia model and no effects in calcium chloride- or aconitine-induced arrhythmias. Moreover, the tested compounds demonstrated therapeutic antiarrhythmic activity, restoring a normal sinus rhythm immediately after the administration of the arrhythmogen adrenaline. Notably, none of the tested derivatives affected the normal electrocardiogram (ECG) parameters in rodents, which excludes their proarrhythmic potential. Finally, all tested compounds decreased blood pressure in normotensive rats and reversed the pressor response to methoxamine, suggesting that their hypotensive mechanism of action is connected with the blockade of α_1_-adrenoceptors. Our results confirm the antiarrhythmic and hypotensive activities of novel arylpiperazine derivatives and encourage their further investigation as model structures for potential drugs.

## 1. Introduction

Various factors in our daily lives, including an unhealthy diet, stress, obesity, tobacco, or even a history of infections, may negatively affect our heart and blood vessel function, leading to the development of cardiovascular diseases. Globally, sudden cardiac death due to heart attack and stroke continues to be one of the most common causes of mortality [1]. These life-threatening conditions are directly connected with heart rhythm disturbances [2,3]. Unfortunately, not only is the prevalence of arrhythmia cases growing, but also its pharmacotherapy remains unsatisfactory [4,5,6,7].

Classic antiarrhythmic drugs are divided according to their mechanisms of action and the Vaughan-Williams classification into four main classes: sodium channel blockers, β-blockers, potassium channel blockers, and calcium-channel blockers [8]. Despite the well-established and common use of antiarrhythmics, many controversies surround their effectiveness and safety [7,9,10,11,12]. The Cochrane review showed that amiodarone remains the most effective in the reduction of sudden cardiac death in comparison to other antiarrhythmics [13]. However, treatment with amiodarone may lead to serious side effects, such as pulmonary fibrosis, hypothyroidism or hyperthyroidism, liver injury, myopathy, and corneal epithelial opacities [14]. Another commonly used antiarrhythmic drug, propafenone, possesses significant proarrhythmic effects and even at therapeutic doses may cause the prolongation of the PR interval, widening the QRS complex and QT interval, leading to ventricular tachycardia or bradycardia [8,15]. β-blockers, often used as first-line antiarrhythmic therapy, may induce not only cardiovascular side effects, i.e., bradycardia or atrioventricular blocks, but also exacerbate asthma or dyslipidemia [8]. Verapamil, a calcium channel blocker indicated in the treatment of atrial fibrillation or atrial flutter, can cause unwanted bradycardia, atrioventricular blocks, peripheral edema, and constipation [8,16]. Among the varied adverse effects, the hazardous proarrhythmic potential of numerous antiarrhythmic drugs requires caution while prescribing [17,18,19,20,21,22,23]. The proarrhythmic action of antiarrhythmics is associated with the blockade of voltage-gated potassium ion channels, mainly Kv11.1 (*hERG*-human ether-a-go-go-related gene) and Kv7.1 (KvLQT1) channels, which are responsible for securing the delayed repolarization of myocardial potential [24]. The blockade of these ion channels within certain limits results in the prolongation of the cardiac action potential, which is important for the antiarrhythmic effect. However, the excessive inhibition of these potassium channels may cause the loss of repolarization reserves and the development of life-threatening *torsades de pointes* [24,25]. On the other hand, an increase in cardiac late sodium current is also responsible for the reduction of repolarization reserves in cardiac myocytes, prompting long QT syndrome and developing arrhythmias [26]. The component of the electrocardiogram (ECG) associated with the duration of all cellular action potentials in the ventricles is the QT interval. Therefore, every prolongation in cardiac action results in the QT interval prolongation, presenting the risk of *torsades de pointes* arrhythmia [24,27]. Thus, one of the main aspects of assessing the cardiovascular safety of novel compounds should be the investigation of their possible effect on the QT interval [27]. Taking all of this into account, there is a huge necessity to develop novel and safer compounds that can not only restore the normal sinus rhythm but also induce other beneficial cardiac effects.

One of the key elements of cardiac function and blood pressure homeostasis are adrenergic receptors, which mediate the actions of the sympathetic nervous system by transducing the external catecholamine stimulus into an intracellular signal [28,29,30,31]. Among them, α_1_-adrenergic receptors may play a role in the pathogenesis of arrhythmias due to their expression in the myocardium [32,33,34] and their mediation in inotropic effects [35]. The involvement of α_1_-adrenergic stimulation in proarrhythmic effects was confirmed in various studies on isolated rat hearts [36,37,38]. Moreover, scientists discovered the potential antiarrhythmic activity of α_1_-adrenolytics [39,40,41,42,43,44]. Among the described active compounds, there are arylpiperazine alkyl derivatives, which demonstrated α_1_-adrenolytic properties and antiarrhythmic activity in rodents [45]. Moreover, some salicylamide derivatives also showed an affinity for α_1_-adrenergic receptors, which may suggest their possible potential to normalize disturbed heart rhythm [46,47,48].

Taking the above into account, we decided to investigate a group of six novel compounds, chemically salicylamide derivatives with arylpiperazine alkyl moiety, and assess their potential cardiovascular effects. We aimed to determine their affinity for α_1_- and α_2_-adrenergic receptors, as well as their antiarrhythmic activity in different rat models of arrhythmia, and the possible hypotensive effects.

## 2. Results

### 2.1. The Tested Compounds Showed High to Moderate Affinity for α_1_-Adrenergic Receptors

We investigated the affinity of arylpiperazine derivatives for α-adrenergic receptors. The radioligand binding assays revealed that all tested compounds possessed high to moderate affinity for α_1_-adrenergic receptors but no affinity for α_2_-adrenergic receptors (Table 1).

### 2.2. The Tested Compounds Showed Prophylactic Antiarrhythmic Activity in the Arrhythmia Model Induced by Adrenaline, but Not by Calcium Chloride or Aconitine in Rats

In the next step of our experiments, we investigated the prophylactic antiarrhythmic properties of the studied compounds in different animal models of arrhythmia. As all tested derivatives possessed high affinity for α_1_-adrenergic receptors, we used adrenaline (20 μg/kg) as an arrhythmogen to induce heart rhythm irregularities in rats. Considering the mechanisms of action of classic antiarrhythmic drugs, we also used calcium chloride (140 mg/kg), acting via cardiac calcium channels, and aconitine (20 μg/kg) acting via sodium channels, to evoke arrhythmia (Table 2).

All the tested compounds, administered at a range of doses 0.02–1 mg/kg before adrenaline, reduced the number of heart rhythm disturbances, such as extrasystoles, conduction blocks, and bradycardia, protecting animals from death. ED_50_ values for each compound are presented in Table 3.

None of the studied compounds was active in calcium chloride- and aconitine-induced models of arrhythmia and did not reduce the number of heart rhythm irregularities in rats.

### 2.3. The Tested Compounds Showed Therapeutic Antiarrhythmic Activity in the Adrenaline-Induced Arrhythmia Model in Rats

The prophylactic antiarrhythmic activity of the studied derivatives in adrenaline-induced models of arrhythmia set the ground for the investigation of their potential therapeutic antiarrhythmic activity in the same arrhythmia model.

All the tested compounds (except for JJGW12) significantly reduced the number of adrenaline-induced extrasystoles by 58–86% compared to the control group (F(6,30) = 10.61, *p* < 0.0001) (Figure 1).

All the tested compounds (except for JJGW12) also reduced the occurrence of bradycardia and atrioventricular blocks in adrenaline-treated rats (Table 4).

### 2.4. The Tested Compounds Did Not Influence the ECG Parameters in Rats (Except for JJGW02 and JJGW11, Which Decreased the Heart Rate Significantly)

In order to exclude potential proarrhythmic effects of the tested derivatives, we investigated their influence on normal ECG in rats.

None of the tested compounds administered at a dose of 5 mg/kg affected PQ interval (JJGW01: (F(4,20) = 0.639, *p* = 0.641), JJGW02: (F(4,20) = 1.280, *p* = 0.311), JJGW03: (F(4,20) = 1.426, *p* = 0.262), JJGW07: (F(4,20) = 1.269, *p* = 0.315), JJGW11: (F(4,20) = 0.370, *p* = 0.827), JJGW12: (F(4,20) = 0.486, *p* = 0.746)). Moreover, studied compounds did not influence QRS complex (JJGW01: (F(4,20) = 0.920, *p* = 0.472), JJGW02: (F(4,20) = 0.846, *p* = 0.513), JJGW03: (F(4,20) = 0.299, *p* = 0.875), JJGW07: (F(4,20) = 0.730, *p* = 0.582), JJGW11: (F(4,20) = 0.815, *p* = 0.530), JJGW12: (F(4,20) = 0.519, *p* = 0.723)) as well as QT_c_ interval (JJGW01: (F(4,20) = 2.345, *p* = 0.090) JJGW02: (F(4,20) = 1.454, *p* = 0.253), JJGW03: (F(4,20) = 1.540, *p* = 0.229), JJGW07: (F(4,20) = 0.783, *p* = 0.549), JJGW11: (F(4,20) = 1.181, *p* = 0.349), JJGW12: (F(4,20) = 0.785, *p* = 0.549)) (Table 5).

JJGW02 administered at a dose of 5mg/kg decreased the heart rate by 10%, 15%, and 16% in the 5th, 10th, and 15th min after the injection, respectively (F(4,20) = 10.580, *p* < 0.0001). Moreover, JJGW11 administered at a dose of 5 mg/kg decreased the heart rate by 8% and 10% in the 10th and 15th min of the ECG recording, respectively (F(4,2) = 8.106, *p* < 0.001). Other compounds did not affect the heart rate in rats during the ECG recording (JJGW01: (F(4,20) = 1.512, *p* = 0.236), JJGW03: (F(4,20) = 0.664, *p* = 0.625), JJGW07: (F(4,20) = 1.425, *p* = 0.262), JJGW12: (F(4,20) = 1.605, *p* = 0.212)) (Table 5).

### 2.5. The Tested Compounds Decreased Blood Pressure in the Normotensive Rats

Considering the high affinity of the tested compounds towards α_1_-adrenergic receptors and the role of these receptors in blood pressure regulation, we evaluated the influence of the studied derivatives on the blood pressure of the normotensive rats.

The administration of JJGW01 at a dose of 1 mg/kg decreased the systolic blood pressure by 9–15% (112.5 mmHg vs. 102.5–95.5 mmHg) between the 5th and 60th min after the injection (Time: F(7,70) = 9.019, *p* < 0.0001, Treatment: F(1,10) = 25.930, *p* < 0.001, Interaction: F(7,70) = 2.072, *p* = 0.058), while decreasing the diastolic blood pressure by 16% and 17% (87.7 mmHg vs. 73.2–73.8 mmHg) in the 40th and 60th min after administration, respectively (Time: F(7,70) = 3.941, *p* < 0.01, Treatment: F(1,10) = 7.603, *p* < 0.05, Interaction: F(7,70) = 1.333, *p* = 0.248) (Table 6).

JJGW02 at a dose of 2.5 mg/kg decreased the systolic blood pressure by 8–10% (114.3 mmHg vs. 105.2–102.8 mmHg) between the 5th and 30th min and between the 50th and 60th min after administration (Time: F(7,70) = 3.091, *p* < 0.01, Treatment: F(1,10) = 14.30, *p* < 0.01, Interaction: F(7,70) = 2.050, *p* = 0.061), as well as decreased the diastolic blood pressure by 13–15% (84.8 mmHg vs. 73.8–72.0 mmHg) between the 5th to 40th min and in the 60th min after administration (Time: F(7,70) = 2.156, *p* < 0.05, Treatment: F(1,10) = 17.52, *p* < 0.01, Interaction: F(7,70) = 1.572, *p* = 0.158) (Table 6).

The hypotensive effect of JJGW03 was observed at a dose of 5 mg/kg—the compound decreased the systolic blood pressure by 16–21% (128.5 mmHg vs.108.0–101.3 mmHg) from the 5th to 60th min after administration (Time: F(7,70) = 13.35, *p* < 0.0001, Treatment: F(1,10) = 31.61, *p* < 0.001, Interaction: F(7,70) = 6.631, *p* < 0.0001), and decreased the diastolic blood pressure in the 5th and 10th min after administration by 18% and 20%, respectively (101.5 mmHg vs. 83.2–81.2 mmHg) (Time: F(7,70) = 4.415, *p* < 0.001, Treatment: F(1,10) = 4.059, *p* = 0.072, Interaction: F(7,70) = 3.942, < 0.01) (Table 6).

On the other hand, JJGW07 at a dose of 5 mg/kg decreased the systolic blood pressure in the rats by 3–13% (117.7 mmHg vs. 113.8–102.8 mmHg) from the 5th to 60th min after administration (Time: F(7,70) = 10.18, *p* < 0.0001, Treatment: F(1,10) = 34.72, *p* < 0.001, Interaction: F(7,70) = 1.295, *p* = 0.266), but did not influence the diastolic blood pressure in the normotensive rats (Time: F(7,70) = 2.471, *p* = 0.025, Treatment: F(1,10) = 21.79, *p* < 0.001, Interaction: F(7,70) = 0.225, *p* = 0.978) (Table 6).

JJGW11 at a dose of 2.5 mg/kg decreased the systolic blood pressure by 7–14% (116.7 mmHg vs. 108.5–100.2 mmHg) between the 5th and 60th min after injection (Time: F(7,70) = 5.167, *p* < 0.0001, Treatment: F(1,10) = 21.56, *p* < 0.001, Interaction: F(7,70) = 1.164, *p* = 0.334), as well as decreased the diastolic blood pressure in the 60th min after administration by 18% (88.7 mmHg vs. 73.0 mmHg) (Time: F(7,70) = 2.638, *p* < 0.05, Treatment: F(1,10) = 6.612, *p* < 0.05, Interaction: F(7,70) = 0.891, *p* = 0.519) (Table 6).

The injection of JJGW12 at a dose of 5 mg/kg decreased the systolic blood pressure in the rats in the 50th min since administration by 13% (122.5 mmHg vs. 107.0 mmHg) (Time: F(7,70) = 7.167, *p* < 0.0001, Treatment: F(1,10) = 4.824, *p* = 0.052, Interaction: F(7,70) = 1.428, *p* = 0.208), but did not influence the diastolic blood pressure (Time: F(7,70) = 1.712, *p* = 0.120, Treatment: F(1,10) = 0.349, *p* = 0.568, Interaction: F(7,70) = 0.687, *p* = 0.683) (Table 6).

### 2.6. The Tested Compounds Reversed the Pressor Effect of Methoxamine in Rats

In order to establish that the hypotensive activity of the studied compounds was a result of their α_1_-adrenolytic properties, we investigated their pressor response to methoxamine, an agonist of α_1_-receptors.

JJGW01, administered at a dose of 1 mg/kg, attenuated the methoxamine pressor response in rats by 81% (t(5) = 7.103, *p* < 0.001). The pressor effect of methoxamine was also abolished after the injection of JJGW02 and JJGW11 at a dose of 2.5 mg/kg by 100% (t(5) = 5.225, *p* < 0.01) and by 99% (t(5) = 5.287, *p* < 0.01), respectively. Moreover, the administration of JJGW03, JJGW07, and JJGW12 at a dose of 5 mg/kg reversed the pressor effect of methoxamine by 99% (t(5) = 15.610, *p* < 0.0001) by 100% (t(5) = 4.294, *p* < 0.01) and by 97% (t(5) = 7.252, *p* < 0.001), respectively (Figure 2).

## 3. Discussion

In the course of our research aiming to develop the potential candidates for the antiarrhythmic and hypotensive drugs, we have discovered that the group of novel arylpiperazine alkyl derivatives of salicylamide showed prophylactic and therapeutic antiarrhythmic properties in rodents. Furthermore, the studied compounds protected animals from death and reduced heart rhythm disturbances in the adrenaline-induced arrhythmia model. Importantly, none of the tested derivatives exposed torsadogenic potential, characteristic of many antiarrhythmic drugs, proving their beneficial safety profile. Additionally, due to antagonistic effects toward α_1_-adrenoceptors, all the compounds decreased systolic and diastolic blood pressure in the normotensive rats.

Adrenergic receptors regulate arterial pressure, blood flow, and cardiac function, presenting the limiting stage of the cardiac response to adrenergic stimulation [52]. Dunaway et al. demonstrated that activation of cardiac α-adrenoceptors reduced cardiac output, ejection fraction, and stroke volume, negatively affecting cardiac function [53]. Correspondingly, Heusch G discovered that the blockade of coronary vascular α-adrenergic receptors might benefit coronary blood flow and myocardial function [54]. Therefore, novel drug candidates targeting these receptors are being developed [42,55,56,57]. Many compounds with arylpiperazine fragments showed a high affinity for α-adrenergic receptors [58,59,60]. Thus, as the first step in our studies, a series of selected compounds were submitted to radioligand binding assays to assess their affinity for α_1_- and α_2_-adrenergic receptors. Our results showed a substantial affinity for α_1_-adrenoceptors (*p*Ki = 7.41–8.40) and no significant interaction with α_2_-adrenergic receptors. Among the tested derivatives, JJGW01 demonstrated the highest affinity for α_1_-adrenergic receptors. Its affinity was even higher than that of phentolamine, the reference compound.

Emotional stress and intense exercise stimulate the sympathetic nervous system, activate cardiac adrenoceptors, and may trigger arrhythmia [61]. A number of studies suggest the role of α_1_-adrenoceptors in antiarrhythmic effects. Knowing that the tested arylpiperazine alkyl derivatives target α_1_-adrenergic receptors, in the following step, we investigated their prophylactic antiarrhythmic activity in the rat adrenaline-induced model of arrhythmia. When adrenaline, used as an arrhythmogen, was bound to the adrenergic receptors, a cascade of cardiac events, such as extrasystoles, conduction blocks, and bradycardia, was triggered. The pretreatment with all the studied compounds reduced the number of post-adrenaline heart rhythm disturbances, indicating their prophylactic antiarrhythmic activity. Significantly, the calculated ED_50_ value for the most active compound, i.e., JJGW07, was 9-fold lower than that of carvedilol, a drug with proven antiarrhythmic properties [62]. The obtained results encouraged further studies on the therapeutic antiarrhythmic activity of the tested compounds in the same rat arrhythmia model. In this experiment, the studied derivatives were coadministered with adrenaline simultaneously to evaluate if they could effectively stop heart rhythm disturbances. Our studies revealed that all the compounds (except for JJGW12) reduced the number of post-adrenaline extrasystoles, conduction blocks, bradycardia, and rodent mortality. As previously mentioned, JJGW07 showed the most robust therapeutic antiarrhythmic activity and outperformed other derivatives. Together, these findings indicate that the studied arylpiperazine alkyl (except for JJGW12) could not only prevent but also treat attacks of arrhythmia induced by adrenaline. All the derivatives showed high to moderate affinity for α_1_-adrenergic receptors. Hence, most likely, the observed pharmacological effects are due to interaction with α_1_-adrenergic receptors or other-not-yet tested targets (e.g., ion channels). Our results suggest that arylpiperazine alkyl derivatives of salicylamide could effectively treat arrhythmia caused by catecholamines. However, more studies are necessary.

As the next step, we investigated the prophylactic antiarrhythmic potential of the studied compounds in rodent models of arrhythmia induced by the administration of either calcium chloride or aconitine. Calcium chloride changes intracellular Ca^2+^ levels, whereas aconitine modifies Na^+^ concentration, causing arrhythmias manifested by extrasystoles, fibrillations, blocks, bradycardia, and increased animal mortality. None of the tested compounds showed activity in these arrhythmia models, so we can assume that Ca^2+^ and Na^+^ channels do not play a significant role in their antiarrhythmic activity.

Increasing evidence suggests that sympathetic nervous system activity is associated with ventricular tachyarrhythmias and sudden death [63]. The potential to induce life-threatening arrhythmias, i.e., *torsade de pointes*, is directly linked with the prolongation of the QT interval in the ECG recordings [51,64]. Therefore, next, the proarrhythmic properties of the tested derivatives were investigated by analyzing their effects on the normal ECG in rats. Notably, none of the studied compounds at the highest tested dose, i.e., 5 mg/kg, affected the QT interval and exposed the proarrhythmic potential. Therefore, if arylpiperazine alkyl derivatives showed no torsadogenic potential at the higher doses, we may assume that they do not possess proarrhythmic properties at lower doses. However, JJGW02 and JJGW11 affected the heart rate and showed negative chronotropic effects, generating a risk of developing bradycardia, but at doses 11-fold and 26-fold higher than the calculated ED_50_ value in the adrenaline-induced arrhythmia model.

For approximately 50 years, drugs targeting α_1_-adrenergic receptors have been used to regulate blood pressure [65]. α_1_-adrenolytics counteract sympathetic dominance, which is considered a pathogenic factor in hypertension [66]. Therefore, as the next step in our study of novel arylpiperazine alkyl derivatives of salicylamide, we evaluated their effect on systolic and diastolic blood pressure in the normotensive rats. All the tested compounds showed significant hypotensive activity after a single intravenous administration, with JJGW01 acting the strongest. We discovered that JJGW01 decreased blood pressure at a dose 10-fold higher than the median antiarrhythmic dose in the adrenaline-induced arrhythmia. On the other hand, JJGW07, which showed the most potent antiarrhythmic effect, lowered blood pressure at a dose 125-fold higher than the ED_50_ value for antiarrhythmic effect. Thus, compounds such as JJGW01 could be a promising therapeutic option for patients suffering from heart rhythm disturbances accompanied by high blood pressure, whereas JJGW07 could be beneficial for patients with arrhythmia alone.

As the last step, we evaluated the effect of the studied compounds on the vasopressor response to methoxamine, an agonist of α_1_-adrenergic receptors [67]. The inhibition of the vasopressor response to the mentioned α_1_-adrenomimetic by the pretreatment with the test derivatives proves that their mechanism of hypotensive action is a result of α_1_-adrenolytic properties. In our study, all studied compounds diminished the hypertensive effect induced by methoxamine, with JJGW03 acting the strongest. Therefore, we can assume that the hypotensive effect of the tested arylpiperazine alkyl derivatives was due to their α_1_-adrenolytic properties.

Our study has some limitations. First, since β-adrenoceptors play an essential role in cardiac function, in future studies using radioligand binding assays, we need to assess the affinity of the tested compounds for β-adrenergic receptors, to reveal their mechanism of antiarrhythmic and hypotensive activity fully. Even though none of the studied derivatives showed effectiveness in the calcium chloride- and aconitine-induced models of arrhythmia, we cannot entirely eliminate the role of Na^+^ and Ca^2+^ in their mechanism of antiarrhythmic action, so it requires further investigation and evaluation of their effects on voltage-gated sodium and calcium channels in radioligand binding studies. Finally, the intrinsic activity toward α_1_-adrenergic receptors needs evaluation using *in vitro* functional assays.

## 4. Materials and Methods

### 4.1. Drugs

The tested compounds 2-{4-[4-(3-chlorophenyl)piperazin-1-ylo]butoxy}benzamide hydrochloride (JJGW01), 2-{5-[4-(3-chlorophenyl)piperazin-1-ylo]pentoxy}benzamide hydrochloride (JJGW02), 2-{6-[4-(3-chlorophenyl)piperazin-1-ylo]hexoxy}benzamide hydrochloride (JJGW03), 2-{4-[4-(2-metoxyphenyl)piperazin-1-ylo]butoxy}benzamide hydrochloride (JJGW07), 2-[(3-{[4-(2-metoxyphenyl)piperazin-1-ylo]methyl}phenyl)metoxy}benzamide hydrochloride (JJGW11), and 2-[(4-{[4-(2-metoxyphenyl)piperazin-1-ylo]methyl}phenyl)metoxy}benzamide hydrochloride (JJGW12) were synthesized in the Department of Organic Chemistry and Technology, Faculty of Chemical and Engineering and Technology, and Cracow University of Technology. The synthesis and biological properties of tested compounds were described earlier by Kowalski et al. [68] and Jaśkowska et al. [69].

The studied compounds were dissolved in saline (0.9% NaCl, Polpharma, Poland) and administered intravenously (*iv*). Chemicals used in the radioligand studies, i.e., phentolamine (Sigma-Aldrich, Darmstadt, Germany) or clonidine (Sigma-Aldrich, Germany), were dissolved in saline. Commercially available reagents such as adrenaline (Polfa S.A., Warszawa, Poland), methoxamine (Sigma-Aldrich, Germany), calcium chloride (Sigma-Aldrich, Germany), and aconitine (Sigma-Aldrich, Germany) were dissolved in saline and administered *iv*. Thiopental (Sandoz GmgH, Vienna, Austria) was also dissolved in saline and administered intraperitoneally (*ip*). Heparin (Polfa S.A., Poland) was used as an anticoagulant during the experiments. The control groups received saline as a vehicle.

### 4.2. Animals

All experiments were performed on male *Wistar* rats, weighing 200–250 g, purchased from an accredited animal facility at the Faculty of Pharmacy, Jagiellonian University Medical College, Krakow, Poland. The animals were housed in groups of 3 rats in plastic cages (42.7 cm × 26.7 cm), in a room with controlled temperature (22 ± 2 °C), appropriate humidity (40–60%), and 12 h light/dark cycle. The standard pellet food and filtered tap water were permanently available. The animals were assigned randomly to either control or treatment groups, and each group consisted of 5–6 Wistar rats. All the injections were administered in 1 mL/kg volume by the trained experimenter blind to the treatments. The animals were used only once in each test and were immediately euthanized after each procedure. The procedures involving animals and their care were conducted according to current European Community and Polish legislation on animal experimentation.

### 4.3. Radioligand Binding Assay

The α_1_- and α_2_-adrenoceptor radioligand binding assays were performed on the rat cerebral cortex using previously described methods [70]. [^3^H]-prazosin (19.5 Ci/mmol, α_1_-adrenoceptor) and [^3^H]-clonidine (70.5 Ci/mmol, α_2_-adrenoceptor) were utilized as specific ligands. The brains were homogenized using the ULTRA-TURRAX homogenizer in 10 mL of an ice-cold 50 mM Tris-HCl buffer (pH 7.6). The homogenates were centrifuged at 20,000× g for 20 min (0–4 °C). Subsequently, the cell pellet was resuspended in the Tris–HCl buffer and centrifuged again. Radioligand binding assays were carried out in plates (MultiScreen/Millipore). The final incubation mixture, with a volume of 300 μL, consisted of 240 μL of the tissue suspension, 30 μL of the radioligand solution, and 30 μL of the buffer containing 7–8 concentrations of the studied compounds. In order to measure the unspecific binding, 10 μM phentolamine (for [^3^H]-prazosin) or 10 μM clonidine (for [^3^H]-clonidine) were utilized. The incubation was completed by rapid filtration through Whatman GF/C filters using a vacuum manifold (Millipore). The filters were then washed twice with the assay buffer and placed in scintillation vials with a liquid scintillation cocktail. Radioactivity was measured in a WALLAC 1409 DSA liquid scintillation counter (Perkin Elmer, USA). All the assays were performed in duplicates, and the inhibitory constants (Ki) were calculated.

### 4.4. Prophylactic Antiarrhythmic Activity in Adrenaline-, Aconitine-, and Calcium Chloride-Induced Arrhythmia

All the experiments were performed according to the method described by Szekeres and Papp [71]. The animals were anesthetized with thiopental (75 mg/kg *ip*). Depending on the model, either adrenaline, aconitine, or calcium chloride was used as an arrhythmogen. Adrenaline and aconitine were administered *iv* at a dose of 20 µg/kg, whereas calcium chloride was administered at a dose of 140 mg/kg to induce heart rhythm disturbances. The tested compounds were injected *iv* 15 min before the arrhythmogen. Aspel ASCARD apparatus (standard II lead, with the tape speed 50 mm/s and voltage calibration 1 mV = 1 cm) was used for ECG measurements. The ECG was recorded during the first 2 min and in the 5th, 10th, and 15th min after the arrhythmogen injection (Figure 3A). The lack or decreased amount of extrasystoles, atrioventricular blocks, bradycardia, and fibrillation in the ECG recordings compared to the control group was the criterion of the antiarrhythmic activity. The ED_50_ was calculated using the method of Litchfield and Wilcoxon [50]. All the tested compounds were administered at a dose of 5 mg/kg, and we gradually decreased the dose by half until the antiarrhythmic activity disappeared.

### 4.5. Therapeutic Antiarrhythmic Activity in Adrenaline-Induced Arrhythmia

The experiments were carried out according to the method described by Szekeres and Papp [71]. The heart rhythm disturbances, such as extrasystoles, conduction blocks, and bradycardia, were induced by *iv* injection of adrenaline (20 µg/kg) to anesthetized rats (thiopental, 75 mg/kg, *ip*). The studied compounds were administered *iv* immediately after the injection of adrenaline, at a dose of 1 mg/kg or 5 mg/kg, depending on the prophylactic activity of the studied compounds and their calculated ED_50_ values (if ED_50_ < 0.3, the dose 1 mg/kg was used—JJGW01, JJGW07, and JJGW11; if ED_50_ > 0.3, the dose 5 mg/kg was used—JJGW02, JJGW03, and JJGW12). The Aspel ASCARD apparatus (standard II lead, with a tape of speed 50 mm/s and voltage calibration of 1 mV = 1 cm) was used for the ECG measurements. The ECG was recorded during the first 2 min and in the 5th, 10th, and 15th min after the adrenaline injection (Figure 3B). The lack or decreased amount of extrasystoles, atrioventricular blocks, and bradycardia in the ECG recordings compared to the control group was the criterion of antiarrhythmic activity [72].

### 4.6. The Effect on a Normal Electrocardiogram in Rats

The procedure was performed to exclude the negative effects of the tested compounds on the normal ECG, according to the method described earlier [62]. The Aspel ASCARD apparatus (standard II lead, with the tape speed of 50 mm/s and voltage calibration of 1 mV = 1cm) was used for the ECG measurements. Firstly, animals were anesthetized with thiopental (75 mg/kg *ip*). The ECG recordings were created prior and in the 5th, 10th, and 15th min after *iv* administration of the tested compounds (Figure 3C). The influence on PQ, QT_c_ interval, QRS complex, and heart rate was evaluated. The Bazzett’s formula: QT_c_ = QT/√RR was used to calculate QT_c_ [51]. All the studied compounds were administered at a dose of 5 mg/kg.

### 4.7. ECG Waveform Analysis

Automated ECG waveform analysis was evaluated as described earlier [55], using Eleven Maze software version 0.1 from Eleven Products Sp. z o.o. (Krakow, Poland), which uses artificial intelligence neural network-based models (e.g., [73]) and statistical and mixture modeling features of ECG signals [74]. The software, under expert supervision, recognized and classified the heart rhythm irregularities. It also calculated PQ, QRS, QTc, and rate parameters.

### 4.8. The Influence on Blood Pressure in the Normotensive Rats

The experiment was performed according to the method described earlier [75]. Normotensive Wistar rats were anesthetized with thiopental (75 mg/kg *ip*). The right carotid artery was cannulated with a polyethylene tube filled with heparin solution, using a Datamax apparatus (Columbus Instruments, Columbus, OH, USA) [71]. After 15 min of the stabilization period, the studied compounds were administered *iv*, and their effect on systolic and diastolic blood pressure was evaluated (Figure 4A). The tested derivatives were administered at a dose of 5 mg/kg, and their dose was gradually decreased until the hypotensive activity disappeared.

### 4.9. The Influence on Blood Vasopressor Response in Rats

The experiment was performed to establish the effect of the studied derivatives on the pressor response to methoxamine (150 µg/kg, *iv*), according to the methods described previously [75]. Normotensive Wistar rats were anesthetized with thiopental (75 mg/kg *ip*). The right carotid artery was cannulated with a polyethylene tube filled with heparin solution using a Datamax apparatus (Columbus Instruments, Columbus, OH, USA). After a 15-min stabilization period, the pressor response to methoxamine before (control) and 5 min after the administration of the tested compounds was measured (Figure 4B). The tested compounds were administered at the lowest hypotensive dose.

### 4.10. Statistic Analysis

The results are presented as either means ± SD (or SEM in case of radioligand studies) or as a number of animals in which specific cardiac events (extrasystoles, fibrillations, bradycardias, or mortality) occurred. In our analysis, we used a paired *t*-test, one-way ANOVA followed by Dunnet’s *post hoc*, and also one-way or two-way repeated measures ANOVA followed by Dunnet’s or Bonferroni’s *post hoc*. *p* < 0.05 was considered significant. All data were statistically evaluated with Prism 9.0 software (GraphPad Software, San Diego, CA, USA).

## 5. Conclusions

Our study demonstrated that selected novel arylpiperazine alkyl derivatives could be an attractive therapeutic option for patients with heart rhythm disturbances accompanied by hypertension. However, further studies are necessary to fully determine the mechanisms involved in the cardiovascular effects of the studied compounds, as well as their safety profiles.

## Figures and Tables

**Figure 1 ijms-24-00293-f001:**
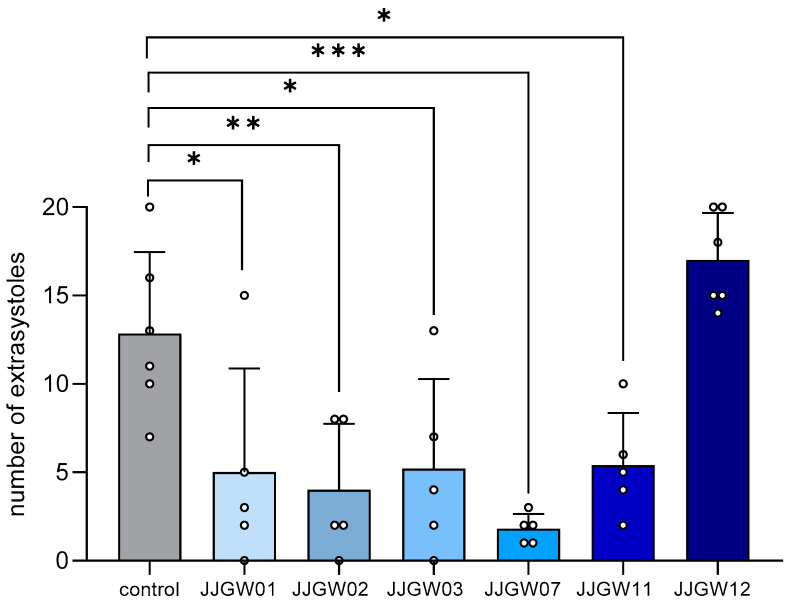
The therapeutic antiarrhythmic activity of the studied compounds in the adrenaline-induced arrhythmia model in rats. After an intravenous (*iv*) injection of adrenaline (20 µg/kg), the tested compounds were immediately administered *iv*. The control group received no additional treatment. The electrocardiogram (ECG) was recorded for the first 2 min and then at the 5th, 10th, and 15th min of the experiment. The criterion of antiarrhythmic activity was the decrease in or complete absence of extrasystoles in the ECG recording compared with the control group. The results are presented as means ± SD for active or maximum tested doses (1 mg/kg for JJGW01, JJGW07, and JJGW11, and 5 mg/kg for JJGW02, JJGW03, and JJGW12). Statistical analysis: one-way ANOVA (Dunnett’s *post hoc*); * *p* < 0.05, ** *p* < 0.01, *** *p* < 0.001; *n* = 5–6 rats.

**Figure 2 ijms-24-00293-f002:**
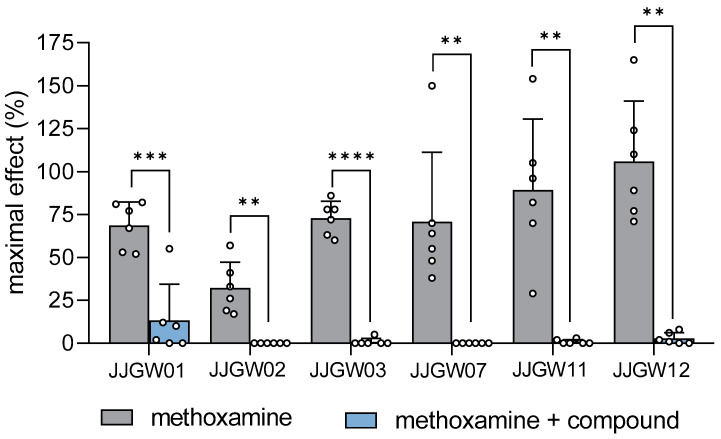
The effect of arylpiperazine derivatives on the blood pressure response to methoxamine in rats. Pressor response to methoxamine (150 μg/kg) was estimated before and 5 min after the intravenous (*iv*) administration of the studied compounds at the lowest hypotensive doses (JJGW01 1mg/kg, JJGW02 2.5 mg/kg, JJGW03 5 mg/kg, JJGW07 5 mg/kg, JJGW11 2.5 mg/kg, JJGW12 5 mg/kg). The results are presented as means ± SD. Statistical analysis: paired *t*-test; ** *p* < 0.01, *** *p* < 0.001, **** *p* < 0.0001; *n* = 6 rats.

**Figure 3 ijms-24-00293-f003:**
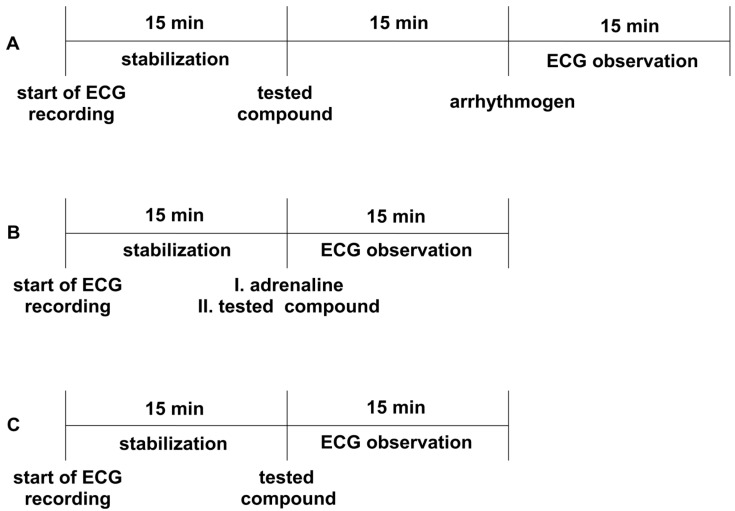
The experimental setup for studying prophylactic (Panel **A**) and therapeutic (Panel **B**) antiarrhythmic activity in rat models of arrhythmia, as well as the influence on electrocardiogram (ECG)_ parameters (Panel **C**). Adrenaline (20 μg/kg), calcium chloride (140 mg/kg), or aconitine (20 μg/kg) were used as arrhythmogens. All the compounds were administered intravenously (*iv*). The ECG observation was during the first 2 min and in the 5th, 10th, and 15th min.

**Figure 4 ijms-24-00293-f004:**
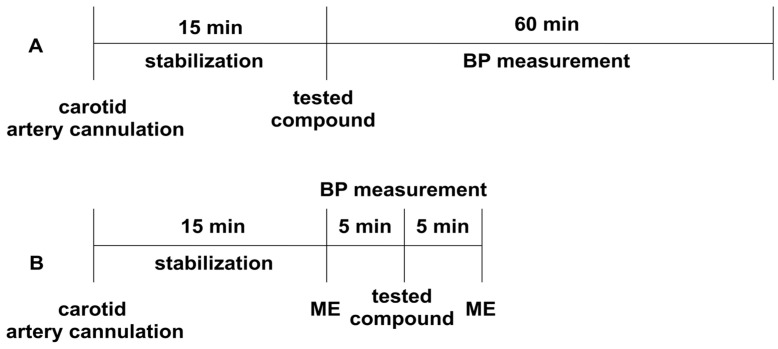
The experimental setup for studying the influence of the tested compounds on blood pressure (Panel **A**), and the pressor response to methoxamine (Panel **B**) in normotensive rats. All the compounds were administered intravenously (*iv*). ME—methoxamine (150 µg/kg), BP—blood pressure.

**Table 1 ijms-24-00293-t001:** The affinity of the tested compounds for α_1_- and α_2_-adrenergic receptors.

Treatment	Adrenergic Receptors—*p*K*_i_*	
	α_1_ ^a^	α_2_ ^b^
JJGW01	8.40 ± 0.01	6.13 ± 0.02
JJGW02	8.33 ± 0.02	6.20 ± 0.02
JJGW03	8.11 ± 0.01	5.76 ± 0.02
JJGW07	7.85 ± 0.01	n.a.
JJGW11	8.07 ± 0.04	n.a.
JJGW12	7.41 ± 0.01	6.11 ± 0.01
Phentolamine	8.05 ± 0.01	-
Clonidine	-	8.30 ± 0.01

Data are represented as *pK_i_,* that is −log*K_i_*_,_ and expressed as means ± SEM from three independent experiments performed in duplicates. Inhibition constants (*K_i_*) were calculated according to the equation of Cheng and Prusoff [49]. Radioligand binding was performed using rat cortex tissue. The affinity values were determined using ^a^ [^3^H]-prazosin or ^b^ [^3^H]-clonidine. n.a. (no affinity)—compound did not bind to the receptor at the concentration 10^−5^ M.

**Table 2 ijms-24-00293-t002:** The effect of arylpiperazine derivatives on the cardiac events occurrence in adrenaline-, calcium chloride-, and aconitine-induced arrhythmia models in rats.

Treatment	Dose (mg/kg)	Fibrillations	Extrasystoles	Bradycardia	Blocks	Mortality
Adrenaline-induced arrhythmia						
Control	-	-	6/6	6/6	6/6	6/6
JJGW01	0.25	-	4/6	2/6	4/6	1/6
	0.1	-	5/6	3/6	5/6	0/6
	0.05	-	6/6	6/6	6/6	4/6
JJGW02	1	-	5/6	4/6	5/6	1/6
	0.5	-	5/6	5/6	6/6	0/6
	0.25	-	5/6	5/6	5/6	3/6
JJGW03	1	-	4/5	1/5	4/5	0/5
	0.5	-	4/5	4/5	5/5	4/5
	0.25	-	5/5	5/5	5/5	4/5
JJGW07	0.1	-	4/6	3/6	3/6	0/6
	0.05	-	5/6	4/6	4/6	0/6
	0.02	-	5/6	6/6	6/6	3/6
JJGW11	0.5	-	3/6	2/6	1/6	1/6
	0.25	-	6/6	0/6	6/6	0/6
	0.1	-	4/5	2/5	4/5	1/5
JJGW12	1	-	3/5	1/5	3/5	0/5
	0.5	-	4/5	3/5	4/5	0/5
	0.25	-	6/6	5/6	6/6	3/6
Calcium chloride-induced arrhythmia						
Control	-	6/6	6/6	6/6	6/6	6/6
JJGW01	5	6/6	6/6	6/6	6/6	6/6
JJGW02	5	4/6	5/6	6/6	5/6	6/6
JJGW03	5	4/6	6/6	6/6	6/6	5/6
JJGW07	5	6/6	6/6	6/6	6/6	6/6
JJGW11	5	6/6	5/6	6/6	6/6	6/6
JJGW12	5	4/6	6/6	6/6	6/6	6/6
Aconitine-induced arrhythmia						
Control	-	6/6	6/6	6/6	6/6	6/6
JJGW01	5	6/6	5/6	6/6	6/6	6/6
JJGW02	5	6/6	6/6	6/6	6/6	6/6
JJGW03	5	6/6	6/6	6/6	6/6	6/6
JJGW07	5	6/6	6/6	6/6	6/6	6/6
JJGW11	5	4/6	5/6	6/6	6/6	6/6
JJGW12	5	5/6	6/6	6/6	5/6	6/6

The tested compounds were administered intravenously (*iv*) 15 min before the experiment. The control group received no treatment except the administration of the arrhythmogen. The electrocardiogram (ECG) recordings were performed for 15 min after the *iv* injection of adrenaline (20 μg/kg), calcium chloride (140 mg/kg), or aconitine (20 μg/kg), i.e., during the first 2 min, at the 5th, 10th, and 15th min. The criterion of antiarrhythmic activity was the decrease in or complete absence of heart rhythm disturbances in the ECG recording compared with the control group. Results are presented as the number of animals in which specific cardiac events (fibrillations, extrasystoles, bradycardia, blocks, or mortality) occurred; *n* = 5–6 rats.

**Table 3 ijms-24-00293-t003:** The prophylactic antiarrhythmic activity of arylpiperazine derivatives in the adrenaline-induced arrhythmia model in rats.

Treatment	ED_50_ (mg/kg)
JJGW01	0.10 (0.03–0.36)
JJGW02	0.45 (0.11–1.77)
JJGW03	0.36 (0.08–1.56)
JJGW07	0.04 (0.01–0.15)
JJGW11	0.19 (0.08–0.45)
JJGW12	0.44 (0.10–1.98)

The tested compounds were administered intravenously (*iv*) 15 min before the experiment. The control group received no treatment except the administration of arrhythmogen. The electrocardiogram (ECG) recordings were performed for 15 min after the *iv* injection of adrenaline (20 μg/kg) during the first 2 min, at the 5th, 10th, and 15th min. The ED_50_ values with confidence limits were calculated according to the methods described by Litchfield and Wilcoxon [50]. Each value was obtained from three experimental groups; *n =* 5–6 rats.

**Table 4 ijms-24-00293-t004:** The therapeutic antiarrhythmic activity of arylpiperazine derivatives in the adrenaline-induced arrhythmia model in rats.

Treatment	Dose (mg/kg)	Bradycardia	Blocks	Mortality
Control	-	6/6	5/6	4/6
JJGW01	1	2/5	3/5	0/5
JJGW02	5	2/6	1/6	1/6
JJGW03	5	1/5	3/5	0/5
JJGW07	1	1/5	2/5	0/5
JJGW11	1	1/5	2/5	0/5
JJGW12	5	6/6	6/6	6/6

The tested compounds were administered intravenously (*iv*) immediately after the injection of adrenaline (20 µg/kg). The control group received no treatment except the administration of arrhythmogen. The electrocardiogram (ECG) recordings were performed for 15 min. Results are presented as the number of animals in which specific cardiac events (bradycardia, conduction blocks, mortality) occurred; *n* = 5–6 rats.

**Table 5 ijms-24-00293-t005:** The influence of studied compounds on the electrocardiogram (ECG) parameters and heart rate in rats.

Treatment	Parameters	Time of Observation (min)			
		0	5	10	15
JJGW01	PQ	42.3 ± 3.4	42.3 ± 2.0	41.3 ± 2.4	41.0 ± 3.3
	QRS	49.3 ± 7.8	46.7 ± 5.6	48.0 ± 4.2	50.3 ± 4.6
	QTc	196.0 ± 14.2	193.7 ± 14.9	186.6 ± 9.0	190.6 ± 14.5
	HR	311.8 ± 11.6	318.0 ± 28.5	324.6 ± 23.1	324.0 ± 23.1
JJGW02	PQ	42.6 ± 2.0	41.7 ± 1.8	42.2 ± 1.6	44.8 ± 5.9
	QRS	36.0 ± 1.1	35.2 ± 1.2	36.1 ± 1.8	35.2 ± 0.7
	QTc	183.3 ± 7.1	175.4 ± 14.0	173.2 ± 15.7	175.1 ± 16.0
	HR	311.1 ± 24.4	278.7 ± 37.0 *	265.0 ± 30.1 ***	262.0 ± 37.3 ***
JJGW03	PQ	41.6 ± 2.3	41.4 ± 1.6	40.2 ± 1.7	40.0 ± 0.5
	QRS	39.6 ± 4.7	39.3 ± 3.9	40.8 ± 7.1	40.4 ± 5.2
	QTc	177.6 ± 12.0	188.7 ± 16.3	174.8 ± 7.3	178.1 ± 15.0
	HR	286.6 ± 45.7	289.1 ± 20.2	279.9 ± 15.7	271.9 ± 18.2
JJGW07	PQ	40.7 ± 2.1	40.0 ± 2.2	40.7 ± 3.7	40.7 ± 1.6
	QRS	47.0 ± 6.3	49.7 ± 7.9	48.7 ± 7.2	48.7 ± 9.2
	QTc	196.5 ± 8.0	190.8 ± 4.2	196.3 ± 9.5	192.4 ± 7.5
	HR	328.7 ± 9.3	341.2 ± 11.5	333.2 ± 20.9	336.3 ± 20.9
JJGW11	PQ	45.8 ± 7.2	47.1 ± 5.7	47.4 ± 5.0	47.6 ± 5.1
	QRS	38.5 ± 3.6	41.3 ± 3.0	42.4 ± 4.3	41.7 ± 5.4
	QTc	188.2 ± 12.3	197.3 ± 18.6	193.6 ± 22.0	184.1 ± 16.9
	HR	281.3 ± 50.8	266.5 ± 58.2	258.8 ± 55.0 **	252.4 ± 50.3 ***
JJGW12	PQ	44.1 ± 6.2	43.6 ± 1.2	45.9 ± 2.4	45.5 ± 3.1
	QRS	41.9 ± 2.1	44.5 ± 3.6	44.2 ± 2.3	45.2 ± 5.9
	QTc	190.8 ± 15.2	194.3 ± 16.8	190.3 ± 25.9	195.8 ± 25.5
	HR	254.6 ± 41.2	256.8 ± 47.1	255.2 ± 53.6	250.2 ± 51.5

The tested compounds were administered intravenously (*iv*), and the observation was performed for 15 min post-injection. The results are presented as means ± SD. Statistical analysis: one-way repeated measures ANOVA (Dunnett’s *post hoc*), * *p* < 0.05, ** *p* < 0.01, and *** *p* < 0.001; *n* = 6 rats. Electrocardiogram (ECG) parameters: PQ interval, QRS complex, QT_c_—calculated QT interval according to Bazzett’s formula: QT_c_ = QT/√RR [51], HR—heart rate.

**Table 6 ijms-24-00293-t006:** The effect of arylpiperazine derivatives on the systolic and diastolic blood pressure in the normotensive rats.

Treatment	Dose (mg/kg)	Blood Pressure	Time of Observation (min)							
			0	5	10	20	30	40	50	60
Control	-	SBP	127.5 ± 8.0	128.7 ± 8.6	126.2 ± 5.9	124.3 ± 5.7	120.5 ± 4.7	120.0 ± 7.0	123.5 ± 7.8	121.3 ± 7.4
	-	DBP	92.2 ± 6.0	94.2 ± 5.7	92.2 ± 5.7	90.3 ± 5.9	88.8 ± 5.0	90.2 ± 11.4	87.3 ± 5.7	90.2 ± 11.9
JJGW01	0.5	SBP	130.7 ± 14.9	119.0 ± 13.4	115.7 ± 13.0	114.7 ± 8.8	114.5 ± 10.2	115.2 ± 12.6	116.8 ± 17.3	113.8 ± 16.1
	1.0		112.5 ± 11.6	102.5 ± 9.5 ^d^	100.3 ± 10.4 ^d^	99.7 ± 11.3 ^c^	98.2 ± 10.4 ^c^	95.7 ± 13.3 ^c^	95.5 ± 11.2 ^d^	96.5 ± 10.6 ^c^
	0.5	DBP	95.3 ± 12.6	88.5 ± 14.3	85.0 ± 13.6	84.8 ± 8.5	84.3 ± 6.9	86.0 ± 10.3	88.3 ± 15.3	86.8 ± 15.5
	1.0		87.7 ± 11.2	80.2 ± 10.8	79.0 ± 11.9	77.5 ± 12.1	75.7 ± 10.6	73.8 ± 14.0 ^a^	72.7 ± 11.6	73.2 ± 11.3 ^a^
JJGW02	1.0	SBP	118.0 ± 10.5	113.7 ± 8.7	111.2 ± 6.4	109.7 ± 9.1	105.7 ± 7.5	107.7 ± 11.5	108.7 ± 12.9	111.0 ± 18.0
	2.5		114.3 ± 15.7	103.2 ± 13.0 ^c^	104.2 ± 13.2 ^b^	102.8 ± 13.4 ^b^	103.7 ± 10.8 ^a^	106.5 ± 10.6	103.2 ± 9.4 ^b^	105.2 ± 9.7 ^a^
	1.0	DBP	87.7 ± 14.9	82.7 ± 16.0	83.3 ± 10.3	80.0 ± 14.1	73.7 ± 10.6	75.0 ± 13.2	73.7 ± 13.9	76.7 ± 19.2
	2.5		84.8 ± 12.1	72.0 ± 9.4 ^c^	72.8 ± 11.6 ^b^	73.0 ± 9.6 ^b^	72.3 ± 8.1 ^a^	73.8 ± 9.4 ^a^	73.7 ± 8.7	73.7 ± 7.9 ^a^
JJGW03	2.5	SBP	123.8 ± 17.4	112.5 ± 13.8	109.8 ± 13.5	108.8 ± 15.6	111.8 ± 20.5	111.2 ± 22.3	107.7 ± 19.7	109.2 ± 21.9
	5.0		128.5 ± 8.6	108.0 ± 6.6 ^d^	104.3 ± 6.3 ^d^	101.3 ± 4.5 ^d^	102.3 ± 6.9 ^d^	104.3 ± 3.9^b^	106.0 ± 7.9 ^c^	107.3 ± 6.1 ^b^
	2.5	DBP	89.3 ± 15.6	83.3 ± 12.8	84.8 ± 14.7	86.0 ± 15.8	85.5 ± 14.7	83.7 ± 15.6	81.2 ± 15.0	79.5 ± 15.4
	5.0		101.5 ± 6.1	83.2 ± 5.0 ^a^	81.2 ± 4.0 ^a^	82.0 ± 4.2	83.5 ± 4.3	85.5 ± 5.8	86.7 ± 6.0	87.5 ± 6.3
JJGW07	2.5	SBP	120.0 ± 9.3	116.5 ± 9.6	112.0 ± 9.7	110.2 ± 10.0	108.0 ± 10.8	106.3 ± 12.9	109.5 ± 9.1	112.0 ± 9.9
	5.0		117.7 ± 2.6	113.8 ± 5.6 ^c^	109.5 ± 5.7^c^	105.2 ± 6.3 ^d^	102.8 ± 6.9 ^d^	104.7 ± 5.7^c^	104.7 ± 5.7 ^d^	104.3 ± 5.6 ^c^
	2.5	DBP	92.7 ± 11.7	89.0 ± 11.9	84.7 ± 12.1	81.7 ± 13.0	80.2 ± 14.4	78.7 ± 16.1	77.5 ± 16.7	76.3 ± 18.0
	5.0		85.3 ± 4.1	88.5 ± 6.2	84.2 ± 6.7	81.2 ± 5.3	79.2 ± 5.3	80.3 ± 2.9	79.8 ± 2.5	80.0 ± 2.1
JJGW11	1.0	SBP	118.2 ± 9.7	116.2 ± 8.5	113.8 ± 9.2	111.5 ± 10.4	111.0 ± 10.8	110.0 ± 11.5	110.3 ± 10.7	110.8 ± 10.6
	2.5		116.7 ± 8.1	108.5 ± 8.7 ^b^	103.3 ± 9.5^c^	102.3 ± 10.0 ^c^	101.3 ± 9.2 ^b^	103.0 ± 10.3 ^a^	103.5 ± 12.9 ^b^	100.2 ± 15.9 ^b^
	1.0	DBP	86.8 ± 7.6	81.3 ± 14.4	82.0 ± 10.2	82.3 ± 8.8	83.0 ± 8.2	82.5 ± 8.4	81.7 ± 8.3	81.5 ± 8.4
	2.5		88.7 ± 10.4	84.7 ± 10.0	81.2 ± 8.6	78.5 ± 10.1	77.3 ± 9.9	79.2 ± 10.5	77.7 ± 12.9	73.0 ± 16.2 ^a^
JJGW12	2.5	SBP	122.2 ± 15.6	120.7 ± 13.8	114.5 ± 10.6	109.7 ± 9.3	106.5 ± 13.1	108.8 ± 11.8	110.0 ± 7.5	110.5 ± 8.1
	5.0		122.5 ± 11.9	120.7 ± 9.8	114.5 ± 9.1	112.8 ± 10.0	111.5 ± 10.4	113.0 ± 11.9	107.0 ± 12.7 ^a^	109.8 ± 12.2
	2.5	DBP	95.0 ± 16.0	94.5 ± 16.5	87.5 ± 23.0	83.5 ± 22.4	84.0 ± 21.4	87.8 ± 16.7	89.3 ± 13.4	89.2 ± 13.9
	5.0		95.7 ± 6.0	87.7 ± 7.0	89.3 ± 8.1	88.5 ± 9.0	87.8 ± 8.8	87.8 ± 8.7	85.8 ± 8.1	86.7 ± 10.7

The blood pressure was measured before and 5, 10, 20, 30, 40, 50, and 60 min after the intravenous (*iv*) administration of the tested compounds or saline (control group). The results are presented as means ± SD. Statistical analysis: two-way repeated measures ANOVA (Bonferroni *post hoc)*; ^a^ *p* < 0.05, ^b^ *p* < 0.01, ^c^ *p* < 0.001, and ^d^ *p* < 0.0001; *n* = 5–6 rats. SBP—systolic blood pressure (mmHg), DBP—diastolic blood pressure (mmHg).

## Data Availability

Data is contained within the article.

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
