# Peer review of "Novel Arylpiperazine Derivatives of Salicylamide with α1-Adrenolytic Properties Showed Antiarrhythmic and Hypotensive Properties in Rats"

_ijms, 2022, doi:10.3390/ijms24010293_

Round 1
Reviewer 1 Report
The paper is well-organised and interesting to read, providing impressive information in each section, which is also sustained with detailed and well-illustrated graphics. I recommend that the authors introduce the references for all methods and extensive the introduction.
Reviewer 2 Report
Comments to Authors (General)
· Some typing errors should be corrected in the manuscript.
· The interpretation of results should be improved.
· The numerical values should be exacted for publication.
Comments to Authors (Specific)
· The introduction shall have to be amended based on discussion with recent research works in the related area of interests.
· The mathematical model for analysis shall have to be mentioned in the respective section especially in materials and methods.
· The authors shall have to discuss on the several values of “p” for analysis.
· The experimental setup shall have to be expressed in appropriate forms.
· The exactness and accuracy of data analysis is very important and many analyses should be done. The authors shall have to discuss on the significant points for outstanding achievement by analysis.
· The direct link between potential to induce life-threatening arrhythmias and prolongation of QT interval in the ECG recordings is very important and the authors shall have to discuss in details.
· The authors shall have to prove the numerical percentage or similar forms for hypotensive effect of the tested arylpiperazine alkyl derivatives due to their α1-adrenolytic properties.
· The authors shall have to give the evaluation methods for overcoming the limitations of the analysis.
· The authors shall have to modify the conclusion of the manuscript.

Reviewer 3 Report
This manuscript describes a preclinical study in which the authors tested the effects of arypiperazine derivatives on rats modeled with α₁-adrenergic receptor agonist or proarrhythmic substances. They found the treatment with arypiperazine derivatives was associated with both arrhythmia-protective and arrhythmia-therapeutic effects and at the same time the self-induced-arrhythmia side effect was limited. The authors also found arypiperazine derivatives can mildly reduce the blood pressure in non-treated models and significantly improve the high blood pressure induced by α₁-adrenergic receptor agonist. The idea of exploring the effect of arypiperazine on arrhythmia and α₁-adrenergic receptor-related hypertension is pretty new. And the design, abundant data and prominent results are decent, all of which brings this study novelty and scientific significance. I listed several major concerns need to be addressed.
1. Did the authors take into account any possible β receptor effects from the arypiperazine derivatives? The β1 receptor antagonists also reduce heart rate and conduction, meanwhile the β2 receptor agonists also induce vasodilation and reduce the blood pressure. In order to attribute the observed effects to α₁-adrenergic receptor, the authors should exclude the possible confounding bias from β receptor.
2. Does the “vehicle” in figure 1 mean control? Please explain the meaning. Also clarify the meaning of PQ (PR interval?), QRS (width?) in the legend of table 5.
3. Please shrink the abstract to make it brief and clean.
Round 2
Reviewer 3 Report
This manuscript describes a preclinical study in which the authors tested the effects of arypiperazine derivatives on rats modeled with α₁-adrenergic receptor agonist or proarrhythmic substances. They found the treatment with arypiperazine derivatives was associated with both arrhythmia-protective and arrhythmia-therapeutic effects and at the same time the self-induced-arrhythmia side effect was limited. The authors also found arypiperazine derivatives can mildly reduce the blood pressure in non-treated models and significantly improve the high blood pressure induced by α₁-adrenergic receptor agonist. The idea of exploring the effect of arypiperazine on arrhythmia and α₁-adrenergic receptor-related hypertension is pretty new. And the design, abundant data and prominent results are decent, all of which brings this study novelty and scientific significance. The authors responded well to my questions and made some revisions. After explanations, the conclusion is less vulnerable. Overall, the manuscript offers abundant data and the tables and figures are logically organized.